# A Systematic Study on Zinc-Related Metabolism in Breast Cancer

**DOI:** 10.3390/nu15071703

**Published:** 2023-03-31

**Authors:** Zheng Qu, Qiang Liu, Xiangyi Kong, Xiangyu Wang, Zhongzhao Wang, Jing Wang, Yi Fang

**Affiliations:** Department of Breast Surgical Oncology, National Cancer Center/National Clinical Research Center for Cancer/Cancer Hospital, Chinese Academy of Medical Sciences and Peking Union Medical College, Beijing 100021, China

**Keywords:** breast cancer, zinc, zinc transporters, cancer therapy, metabolism

## Abstract

Breast cancer has become the most common cancer worldwide. Despite the major advances made in the past few decades in the treatment of breast cancer using a combination of chemotherapy, endocrine therapy, and immunotherapy, the genesis, treatment, recurrence, and metastasis of this disease continue to pose significant difficulties. New treatment approaches are therefore urgently required. Zinc is an important trace element that is involved in regulating various enzymatic, metabolic, and cellular processes in the human body. Several studies have shown that abnormal zinc homeostasis can lead to the onset and progression of various diseases, including breast cancer. This review highlights the role played by zinc transporters in pathogenesis, apoptosis, signal transduction, and potential clinical applications in breast cancer. Additionally, the translation of the clinical applications of zinc and associated molecules in breast cancer, as well as the recent developments in the zinc-related drug targets for breast cancer treatment, is discussed. These developments offer novel insights into understanding the concepts and approaches that could be used for the diagnosis and management of breast cancer.

## 1. Introduction

Today, breast cancer is the most common type of cancer in the world and the primary factor that leads to cancer-related deaths among women [1]. There are three primary subtypes of breast cancer: triple-negative breast cancer (15%, where the tumors lack all three molecular markers), ERBB2-positive breast cancer (15–20%), and hormone-receptor-positive/human epidermal growth factor 2 (ERBB2)-negative (70% of patients) breast cancer [2]. In recent years, surgery, chemotherapy, radiotherapy, endocrine therapy, and immunotherapy have significantly improved the prognosis of breast cancer [3]. Despite decades of laboratory, epidemiological, and clinical research, there has been a considerable increase in the incidence of breast cancer [4]. According to recent studies, the probability of breast cancer recurrence can last for 10 to 32 years [5]. Breast cancer remains a major health concern affecting all humankind.

Zinc is a necessary trace element that is needed for gene regulation, enzyme activity, and protein structure. More than 300 enzymes that participate in numerous metabolic and cellular processes in the human body require zinc as a cofactor for exerting their enzymatic activities. Zinc is essential for maintaining the structural stability and DNA-binding activity of more than 2500 transcription factors [6,7]. Hence, normal cellular activity is regulated by zinc metabolism and homeostasis. Zn acts as an intracellular secondary messenger and is necessary for cell cycle regulation and cell proliferation, differentiation, and apoptosis [8,9].

Zinc homeostasis can be maintained in the body by coordinating the activities of metallothioneins (MTs), zinc transporters, and metal-regulated transcription factor 1 (MTF-1) [10]. Zn ions are transported in opposite directions through intracellular and cellular membranes by two primary families of Zn transporters, i.e., SLC30s/ZnTs and SLC39s/ZIPs. The 14 members of the ZIP family (ZIP1 to ZIP14) transport zinc ions from intracellular organelles or extracellular space into the cytosol. The 10 members of the ZnT family (ZnT1 to ZnT10) are responsible for transporting Zn ions from the cytosol to extracellular or intracellular organelles [11] (Figure 1A). ZnT1 is the only SLC30 member that is commonly expressed at the plasma membrane [12], whereas other ZnT transporters are predominantly localized on secretory vesicles or the Golgi apparatus. Most ZIP transporters localize to the plasma membrane, and their cell surface localization increases under zinc-deficient conditions, with the exception of ZIP5. ZIP and ZnT transporters work in opposite but coordinated ways to maintain cellular zinc homeostasis [13]. So far, there is no direct structural information on any human ZnT or ZIP transporters. However, X-ray crystal and cryo-electron microscopy structures of the bacterial ZnT homologue YiiP help to reveal pathway mechanisms and structural information for zinc transport [14,15]. At present, there is no clear study demonstrating that the tissue expression of ZnT family members is directly related to the genesis and development of breast cancer, so this paper mainly elaborates the current research status of some members of the ZIP family related to breast cancer. In addition to zinc transporters, cytoplasmic metallothioneins are involved in zinc chelation and play critical roles in cellular zinc homeostasis. Several proteins that mediate DNA damage and DNA repair mechanisms require zinc [16]. For instance, p53 is seen to be a vital Zn-containing transcription factor that is linked to many cellular responses, such as DNA damage. The integrity of damaged DNA also increases the risk of cancer onset and progression. Zinc deficiency leads to immune system dysfunction and impairs the cellular- and antibody-mediated immune responses [17].

The impairment of cellular zinc homeostasis is associated with diseases such as breast cancer. Changes in serum Zn levels were observed in the malignant tissues of patients with cancer, where epidemiological evidence showed that serum Zn levels were significantly decreased in the majority of cancers, such as breast, ovarian, gastrointestinal, lung, thyroid, and esophageal cancers [18,19,20]. Contrasting results were reported in other studies, which showed that breast cancer tissues accumulated very high levels of Zn ions (significantly higher than normal epithelial breast tissues) [21,22]. Vogel-González et al. assessed different metastatic markers and observed that Zn concentration was especially essential for regulating the microenvironment of triple-negative breast cancer brain metastatic cells, which also improved SerpinB2 expression [23]. SerpinB2 was reported to allow malignant cells to evade the body’s defense system. The above study showed that zinc was essential for the transformational processes undertaken by malignant cells to become tumorigenic and niche-specific. Sullivan et al. carried out a zinc isotope analysis and found that breast cancer tissues showed a mean zinc concentration of 13.3 µg × g^−1^, which was higher than that of benign tumors (*p* < 0.0001) [24]. The accumulation of zinc ions in the tumor tissues was related to the elevated expression levels of cellular zinc import proteins, which suggested that the malignant cells used common mechanisms to selectively increase zinc uptake. Similarly, the expression levels of zinc transporters in the tumors were associated with their malignancy, which indicated that any changes occurring in zinc homeostasis could alter cancer severity.

Recently, several researchers have highlighted the relationship between breast cancer and zinc levels [10,24,25,26]. They noted that abnormal zinc homeostasis can lead to the occurrence and progression of various diseases, including breast cancer. This review summarizes the recent advances that were observed in zinc-related metabolism and breast cancer treatment, and it also outlines the potential clinical applications of zinc ions in preventing, diagnosing, and treating breast cancer.

## 2. Methodology

This study aimed to describe the limitations of existing research, suggest avenues for future research and therapeutic strategies to address gaps in the current literature, and provide practical recommendations for zinc-related metabolism in breast cancer. For this study, a search of the literature was conducted to identify reviews, randomized controlled trials, systematic reviews, meta-analyses, and articles that investigated zinc and breast cancer (e.g., zinc homeostasis, zinc transporters, zinc isotopes, zinc-related treatment, and targets for breast cancer). The electronic databases PubMed, Web of Science, and Google Scholar were searched for studies published from January 1995 to December 2022. The types of studies include human clinical studies, human cell studies, and animal experimental studies. This research is mainly focused on breast cancer, but it also covers some research on other cancers, including lung cancer, liver cancer, and thyroid cancer.

## 3. Zinc Transporters and Breast Cancer

Tissue-specific zinc transporters regulate intracellular and extracellular zinc levels and distribution [11,27]. Multiple zinc transporters collectively help in maintaining zinc homeostasis in mammary epithelial cells [28]. ZIP6 was identified during genetic screening for the estrogen response factors in breast cancer tissues and was regarded as an estrogen-regulated gene, which was positively linked to the estrogen receptor (ER) [26]. Therefore, it has been highlighted that a high ZIP6 expression is a reliable marker of luminal A subtypes in breast cancer [29]. However, ZIP6 expression plays a significant role in the biological functioning of breast cancer cells and acts as a significant predictor of tumor grade, size, and stage. Epithelial–mesenchymal transition (EMT) regulation in breast cancer may be mediated by ZIP6 [30]. Matsui et al. showed that high-glucose environments increased MCF-7 cell viability under hypoxia. Furthermore, they observed that high glucose levels blocked cell apoptosis and promoted the EMT process, which could be attributed to the effects of Zn and ZIP6 [31]. Alterations in zinc and ZIP6 levels in glucose-exposed MCF-7 cells can also enhance their ability to resist a hypoxic environment. In an earlier study, the researchers noted that, when STAT3 transactivated Zip6 expression during zebrafish gastrulation, Snail was localized in the nucleus, which, in turn, inhibited the production of cdh1 (epithelial cadherin (E-cadherin)) and promoted cell migration [32]. Hogstrand et al. showed that the zinc transporter ZIP6 is transcriptionally stimulated by STAT3 and is activated by N-terminal cleavage, which causes ZIP6 plasma membrane localization and Zn influx. This zinc influx triggers Snail activation, which is localized in the nucleus and acts like a transcriptional repressor of epithelial cadherin (E-cadherin), leading to cell dissociation. Furthermore, this implies that breast cancer metastasis [33] is mediated in a ZIP6-dependent manner.

Additionally, ZIP10 aids in the advancement of breast cancer. Kagara et al. revealed an association between ZIP10, cell infiltration, and the metastasis of human breast cancer cells, and they used the real-time quantitative PCR technique to confirm that the breast cancer tissues in patients with lymph node metastasis showed a significantly higher ZIP10 mRNA expression level than the breast cancer tissues in those without metastasis. In contrast, cell migration and zinc uptake were attenuated by ZIP10 knockdown in metastatic cell lines [22]. Additionally, ZIP10 and ZIP6 can combine to form functional heteromeric complexes that interact with neural cell adhesion molecule 1 (NCAM1) to regulate the EMT process [34]. Nakase et al. noted that the lack of ZIP6 or ZIP10 and intracellular Zn^2+^ levels remarkably attenuated the ability of MCF-7 cells to migrate in a 25 mM glucose medium. This finding suggested that Zn transport via ZIP6 and ZIP10 plays a key role in enhancing the motility of MCF-7 cells at high glucose levels. This raises the possibility that ZIP10 could be used as a prospective marker for the metastatic spread of hyperglycemia in breast cancer [35]. Recently, Li et al. noted that the activation of the ITGA10-mediated PI3K/AKT pathway led to osteosarcoma proliferation and chemoresistance [36]. Further research is still needed to determine the role played by ZIP10 in cancer treatment.

ZIP7 regulates the HER2, epidermal growth factor receptor (EGFR), Src, and IGF1R signaling-related cell growth and differentiation pathways, and it also helps in maintaining the intracellular Zn balance [27,37] (Figure 1B). The endoplasmic reticulum (ER) membrane contains ZIP7, whose activity is dependent on the phosphorylation of casein kinase II (CK2). In their study, Taylor et al. observed that the phosphorylation of CK2 causes the Zn gate in the ER to release Zn metal ions from intracellular reserves. These metal ions activate the AKT and ERK tyrosine kinases, which further increase the proliferation of cancer cells [38]. Since over 70% of breast tumors are estrogen-receptor-positive, ZIP7 could serve as a viable biological target for chemotherapy against breast cancer [38,39]. Furthermore, Taylor et al. used three different kinds of phosphoprotein antibody arrays to examine the various kinases that were phosphorylated in response to the ZIP7-mediated secretion of zinc ions, which confirmed that the MAPK, PI3K-AKT, and mTOR signaling pathways were activated as the primary downstream targets of ZIP7 [40]. They further observed that tamoxifen-resistant breast cancer cells showed a higher ZIP7 hyperactivation level. The findings indicated that activated ZIP7 could be used as a probable biomarker for acquired anti-breast cancer antihormones [41]. In addition, Zhang et al. recently identified the upregulation of ZIP7 that led to the suppression of mitophagy as a crucial aspect of myocardial reperfusion injury. They noted that ZIP7 inhibition could help in the treatment of myocardial reperfusion injuries [42].

According to Thomas et al., testosterone promoted cell apoptosis via the androgen signaling and zinc transporter activities of ZIP9 [43]. Since ZIP9 was found to be expressed at a high level in prostate and breast cancer, it could be used as a probable therapeutic target for treating prostate and breast cancer. ZIP9 is considered a novel membrane androgen receptor, which controls apoptosis and regulates zinc homeostasis and associated hormone functions. ZIP9 mediates testosterone to promote apoptosis via MAP kinase and zinc-dependent pathways [44]. In addition, besides zinc transporters, evidence shows that MTs are related to breast cancer progression [45]. MT overexpression is associated with chemoresistance in patients receiving adjuvant therapy after surgery and promotes breast cancer cell invasion by increasing the expression of matrix metalloproteinase-9 (MMP9) [45,46]. Kmiecik et al. showed that MT3 may regulate the invasiveness of breast cancer cells by regulating MMP3 expression, suggesting that MT3 expression may be a potential marker of poor prognosis in triple-negative breast cancer (TNBC) [47]. Chandler et al. assessed the variations occurring in the expressions of Zn transporters and MTs in breast cancer cell lines and noted that MT was only overexpressed in basal-like MDA-MB-231 cells (p53 mutant and ER-) instead of T47D cells (p53 mutant and ER+). The findings suggested that the subtype-specific dysregulation of zinc management could be responsible for the phenotypic properties of breast cancer. They also offered a novel perspective regarding the detection and treatment of various malignant breast cancer phenotypes [48].

In recent years, an increasing number of studies have shown that autophagy is related to the onset and development of breast cancer, and it further promotes tamoxifen resistance in patients with breast cancer [49,50,51]. Several studies have shown that zinc transporters, such as ZIP4, ZIP8, ZnT4, and ZnT10, affect the autophagic process by controlling the entry and exit of zinc ions in autolysosomes [52,53], indicating the essential role of zinc in autophagy regulation [54]. In a recent study, Qi et al. found that MCOLN1/TRPML1, which is a lysosomal non-selective cation channel, controlled the oncogenic autophagy process in cancer via the mechanism of regulating the influx of Zn ions into the cytoplasm [55], indicating the potential for determining the role exerted by Zn ions in the autophagic mechanism of cancer.

## 4. Correlation between Serum Zinc Levels and Breast Cancer

In recent years, numerous studies have determined the role exerted by serum zinc levels in diagnosing and evaluating the prognosis of many diseases. The results have indicated that serum Zn levels may be associated with diseases such as obesity, heart failure, gestational diabetes, and tumors [56,57,58,59,60]. In a recent case–control study, the researchers noted that the serum zinc levels in patients with breast cancer were significantly lower than those in healthy controls [61]. Riesop et al. analyzed tissue samples collected from patients with breast cancer and healthy patients, using the laser ablation–inductively coupled plasma–mass spectrometry (LA-ICP-MS) technique, and they observed that the Zn ion levels in the breast cancer tissues were two times higher than those in the control samples [62]. Bobrowska-Korczak et al. showed that the dietary intake of Zn nanoparticles (ZnNPs) effectively inhibited tumor growth in breast cancer rat models. Thus, Zn supplementation could be regarded as a potential chemotherapeutic approach, which inhibited the progression of breast cancer [63]. Rosa et al. noted a significant decrease in the serum concentrations of retinol, β-carotene, and zinc with an increase in breast cancer stages, and a combined therapy could improve the effectiveness of treatment and enhance the therapeutic effects in patients with breast cancer by decreasing the negative effects of radiation therapy [64]. Although several studies have confirmed that serum Zn levels could be considered a biomarker for breast cancer, most of them are limited to qualitative diagnoses. Thus, it becomes difficult to determine the malignancy or clinical grade of breast cancer based on serum or tissue Zn levels.

## 5. Copper/Zinc Ratio and Breast Cancer

The correlation between serum Zn and/or copper (Cu) levels and breast cancer was also investigated. Cells convert free radical superoxide into hydrogen peroxide and molecular oxygen through redox reactions. Cu ions were found to be a catalytic cofactor in the Cu/Zn superoxide dismutase I (SOD1) enzyme, which is involved in antioxidant defense. The abnormal expression of SOD1 is linked to several cancers [65,66]. Pala et al. conducted a prospective study and observed that elevated blood Cu/Zn levels could serve as an early marker and a crucial risk factor for breast cancer development [67]. According to a meta-analysis study, the serum Cu levels in patients with breast cancer were considerably higher than those in healthy controls and women with benign breast disease. The results also indicated that elevated serum copper and copper/zinc levels and lower zinc levels may be linked to a higher risk of developing breast cancer [68].

## 6. Zinc Isotopes in Breast Cancer Detection and Prognosis Prediction

Zinc (Zn) has five stable isotopes, usually in the divalent (Zn^2+^) form. ^64^Zn and ^66^Zn are seen to be the most abundant isotopes, accounting for 48.63% and 27.90% of total zinc, respectively [69] (Figure 2). Larner et al. estimated the total Zn levels and the isotopic composition in the blood and serum of healthy controls and patients with breast cancer, and they expressed the isotopic composition as a ^66^Zn/^64^Zn ratio. Their results showed that the malignant tumors showed a significantly lighter Zn composition than the blood, body fluids, and healthy breast tissues obtained from other groups. The metal atom brightness in tumor growth indicated that sulfur-rich metallothioneins rather than Zn-specific proteins were predominant in the atom properties of the breast tissue cells [70]. This difference in intrinsic atom compositions may result in the identification of a distinctive early biomarker for cancer. However, the factors that cause breast cancer tissue to have a lighter Zn isotope composition (δ66Zn) than healthy tissue are not known. The MDA-MB-231 cell line for TNBC was used by Schiling et al. with the help of a trickle reactor to compare the Zn isotope fractionations for cellular uptake (Δ^66^Zn uptake) and cellular egress (Δ^66^Zn egress) with those in an in vivo environment. The study observed isotopic fractionation results that contrasted with those noted in the in vivo breast cancer tissue. The observed variations in the Zn isotopic fractionations between the in vitro experiments using the MDA-MB-231 cell line and the in vivo mammary tissues could be attributed to variations in Zn transporter concentrations or intercellular Zn storage mechanisms, such as variations in metallothionein expression [71]. To validate the difference between the in vitro and in vivo zinc isotope fractionations, additional studies utilizing other human breast cancer cell lines with various Zn protein properties are required. Schiling et al. also analyzed the Zn isotopes in the urine samples of patients with cancer. They used a non-invasive test for tracking dysregulated Zn homeostasis caused by malignant tumors. Higher urinary [Zn] and lower urinary δ^66^Zn levels were noted in patients with pancreatic cancer and in patients with prostate cancer than in healthy controls. However, the urinary (Zn) levels in patients with breast cancer were similar to those in healthy control patients [72], which showed that the δ^66^Zn levels in urine samples cannot be used as a reliable prognostic tool for breast cancer.

## 7. Dietary Zinc Intake and Breast Cancer

Zinc is a vital trace element that is involved in many biological processes; however, it cannot be stored in the body. Most zinc (85%) is found in the skeletal muscle and bones of the human body. A very small amount (about 0.1–1% of the total) is in plasma [73]. Organic forms of zinc are usually better absorbed, and foods of animal origin (especially oysters, meat (beef, pork, and mutton), and some seafood), nuts, seeds, and dried beans are good sources of dietary zinc. Vegetables, fruits, starchy tuberous roots, and tubers have a low zinc content [74,75]. Therefore, the dietary intake of Zn metal ions is particularly important. An inadequate zinc intake can be a serious global public health problem, compromising the health of millions of adults and children [76]. The Lancet series on mothers and undernourished children concluded that zinc deficiency accounts for approximately 4% of child mortality and disability-adjusted life years [77]. Several studies have shown that dietary Zn intake may be negatively related to the risk of various diseases, including depression, cognitive impairment, and chronic kidney disease, and a decreased immune defense [7,78,79,80]. Meanwhile, a reduced dietary zinc intake may be positively associated with the progression of type 2 diabetes, atherosclerosis, and metabolic syndrome (MS) [81,82,83]. However, the correlation between zinc intake and cancer is inconclusive. A prospective study carried out in Jiangsu Province, China, with a 10-year follow-up, showed that zinc intake was positively associated with all-cause mortality and cancer mortality [84]. In their study, Epstein et al. noted that a high dietary Zn intake was associated with reduced prostate-cancer-specific mortality after diagnosis [85]. Gutiérrez-González et al. reported that a high Zn dietary intake may increase the risk of low-grade prostate cancer [86]. A meta-analysis showed that dietary zinc intake significantly reduced the risk of pancreatic cancer in the US population [87]. Luo et al. conducted a case–control study and observed that the intake of Zn ions was not linked to colorectal cancer risk [88]. A German study investigated the relationship between breast cancer incidence and the intake of vegetables, fruits, and micronutrients and found that a high zinc intake may reduce breast cancer risk [89]. Furthermore, no clear relationship was noted between dietary Zn intake and the overall breast cancer risk or the risk of various breast cancer subgroups [90]. However, in terms of survival rate, Bengtsson et al. observed that a moderate/high Zn intake could help in increasing the survival rate of patients with breast cancer, based on the Malmö Diet and Cancer Study in Sweden [25]. Pan and colleagues from Canada studied 2322 patients to assess the association between antioxidants and breast cancer risk and found that the supplementation of zinc for 10 years or longer was associated with a statistically significant reduction in premenopausal breast cancer risk, though the overall effect of the total dose or intake from both diet and supplements could not be determined [91]. Conversely, a cohort study from Spain involving 9983 participants reported that there was no evidence for a consistent association between the intake of zinc and breast cancer risk either in overall or premenopausal women [92]. The correlation between dietary Zn intake and breast cancer morbidity and mortality still needs to be corroborated by further research.

## 8. Zinc-Related Drugs and Targets and Breast Cancer

Breast cancer treatment has undergone considerable advancements in recent years owing to the increased number of zinc-related medicines and targets. According to Na et al., zinc metallochaperones (ZMCs), a novel anticancer medication, might preferentially reactivate zinc-deficient p53 mutants by restoring their zinc-binding ability, which facilitated the treatment of patients with BRCA1-deficient breast cancer in a preclinical setting [93]. According to Seattle Genetics, the LIV-1 (SLC39A6) expression level was maintained after hormone therapy at both primary and metastatic sites; however, it was elevated in patients with triple-negative breast cancer. Additionally, a novel LIV-1-targeting antibody–drug combination called SGN-LIV1A was developed to treat metastatic breast cancer. Antibody–drug conjugates (ADCs) are a novel treatment strategy that uses the high specificity displayed by monoclonal antibodies in delivering strong cytotoxic drugs. A new therapy option for patients with metastatic breast cancer and patients with LIV-1-positive breast cancer may emerge as per the findings of the SGN-LIV1A clinical trials [94]. Many clinical trials are currently being conducted, including NCT01969643, a phase I trial wherein patients with metastatic breast cancer are administered SGN-LIV1A monotherapy and the dose expansion phase of SGN-LIV1A in conjunction with Trastuzumab [95], and NCT03310957, a phase Ib/II trial that involves patients with metastatic TNBC who are administered SGN-LIV1A and Pembrolizumab [96]. SGN-LIV1A showed a high response rate in patients with breast cancer and could improve the success rate of neoadjuvant therapy and the recurrence-free survival period. In addition to LIV-1, ZIP7 may also serve as a potential target for breast cancer therapy. ZIP7 is the only known member of the ZIP family that is located on the ER membrane. Under conditions of high ER stress or dysregulated zinc levels, ZIP7 may act as a therapeutic target, because Woodruff et al. observed that a ZIP7 inhibitor and some other small-molecule drugs have the ability to rescue the elevated ER stress in ZIP7 and lower cell proliferation in a Zn-independent manner [97]. Since Notch receptors are thought to exert an oncogenic role in a variety of human malignancies, Nolin et al. observed that ZIP7 could play a vital role in regulating Notch trafficking and signaling. The NVS-ZP7-4 compound could disrupt Notch signaling and cause ER stress and apoptosis [98]. By increasing ER-stress-induced apoptosis in cancer cells, this novel ZIP7 inhibitor could develop into a potential therapeutic agent for treating patients with cancer with high ZIP7 expression levels. Anzilotti et al. also reported the presence of a ZIP7 inhibitor that directly interacted with ZIP7. It disrupted the Notch trafficking pathway, induced apoptosis via ER stress, and altered the Zn levels in ER [99].

## 9. Zinc Oxide Nanoparticles and Breast Cancer

In the past few years, the application of nanotechnology has helped in the development of many nano-drug delivery systems, which show a high selectivity for tumors, do not damage healthy cells or tissues, have a high drug-loading capacity, and can be regarded as an ideal choice for cancer treatment [100,101]. Zinc oxide nanoparticles (ZnONPs) have been widely used in biomedicine because of their high tumor specificity, biocompatibility, and low toxicity [102,103,104] (Figure 3). Mahdizadeh et al. observed that ZnONPs showed strong apoptosis-inducing effects in human breast cancer cell lines (MCF7) and rodent cell lines (TUBO cell lines and cancer models), which highlighted the anti-breast cancer potential of ZnONPs [105]. Vimala et al. synthesized a drug delivery system based on Doxorubicin (DOX) and folic-acid-functionalized polyethylene glycol-coated zinc oxide nanosheets (FA-PEG-ZnO NSs), which increased breast cancer cell killing rates and reduced systemic toxicity in in vitro and mouse assays [106]. Ruenraroengsak et al. used ZnONPs combined with anti-FZD-7 antibodies to target and kill highly expressing FZD-7 breast cancer cells, highlighting the potential clinical application value of this nanoplatform [107]. Recently, Sjs et al. combined Thymoquinone (TQ) with ZnONPs to yield TQ-ZnO nanoparticles that inhibited breast cancer cell proliferation, increased DNA damage, and led to apoptosis in the TNBC cell line MDA-MB-231 [108]. In addition to chemotherapy and targeted immunotherapy, ZnO was used for the synthesis of chitosan-ZnO biological nanocomposites (CS-ZnO BNCs) as radiosensitizers to enhance the effect of radiation therapy in breast cancer models, reflecting advantages in radiation therapy [109].

## 10. Discussion

The control of cellular zinc homeostasis is an extremely complex biological process. At least 14 zinc importers (ZIPs) and 10 zinc exporters (ZnTs), about a dozen metallothioneins (MTs), and a zinc-sensing transcription factor (metal-response element (MRE)-binding transcription factor-1 (MTF-1)) are involved in controlling cellular zinc [11]. Thus, the homeostasis of the free zinc ion concentration in the cytosol and organelles can be further controlled. The most important aspects of homeostatic control are the tight binding of zinc to proteins and the consequent very low cytosolic free zinc ion concentration. Professor Maret and his team have made great contributions and conducted impressive research in this field. Cellular metal ion concentrations are buffered not only by ligand binding but also by transport processes that increase or decrease the metal ion concentrations in specific compartments, a process called muffling [110]. For example, if a cell is challenged by a zinc ion influx, the muffling response will suppress the resulting rise in the cytosolic zinc ion concentration and ultimately restore the cytosolic zinc ion concentration to its original value by shuttling zinc ions into subcellular storage or by removing zinc from the cell [111]. There are at least three pathways for the transient release of zinc ions within and from cells. The proteins involved in homeostasis control (ZIP, ZnT, metallothionein, and MTF-1) restore homeostasis after transients have occurred. The proteins involved in zinc homeostasis control act as mufflers (ZIP and ZnT) and buffers (MT) to restore homeostasis [112]. Remarkably, metallothioneins have critical roles in intracellular zinc homeostasis and buffering reactions due to their zinc binding and transport properties, as well as being a source of signaling zinc ions under redox signaling conditions [111,113]. In addition to the available zinc pools and metallothioneins, a number of studies in recent years have suggested that chaperone-mediated zinc ion transfer now becomes the third known mechanism for allocating zinc ions [114,115,116].

Many recent Zn-related studies have garnered scientific attention. In 2021, Chen et al. observed that Zn metal ions were essential for ferroptosis in renal and breast cancer cells. They also defined ferroptosis as a novel form of regulatory cell death that resulted in significant lipid, peroxidation, and membrane damage [117,118]. Chen et al. noted that ferroptosis sensitivity was also significantly affected by Zn levels, where ZIP7 promoted an increase in cytosolic Zn levels, while ZIP7 inhibition triggered ER stress responses, induced HERPUD1 expression, and protected cells against ferroptosis [119]. More than 2% of the human genome’s sequence is devoted to encoding zinc finger proteins, which are considered the biggest family of transcription factors [120,121]. Scientific evidence has indicated that these proteins are involved in cancer progression. Several studies have shown that zinc finger proteins (ZFPs) are involved in EMT, intratumoral angiogenesis, cell proliferation, and migration in breast cancer [122,123,124,125,126]. Jiang et al. reported that the induction of aerobic glycolysis by the zinc finger E-box binding homeobox 1 (ZEB1) contributes to M2-like tumor-associated macrophage polarization, eventually leading to the growth, metastasis, and chemoresistance of breast cancer cells. Thus, ZEB1 could be used as a target for treating advanced breast cancer [127]. Proteins that bind to precisely defined DNA sequences can be designed using knowledge of the mechanism of zinc finger binding to DNA.

Matrix metalloproteinases (MMPs) belong to a subfamily of Zn-dependent matrix metalloproteinases, and the dysregulation of MMP function is involved in tumor invasion and metastasis. Furthermore, their activity is significantly dependent on the binding of Zn ions to the catalytic site in enzyme molecules. Earlier studies have indicated that MMP activation induces EMT and promotes breast cancer cell motility and progression [128,129]. According to Nazir et al., MMP-9 enhances breast cancer invasion via the Ets-1 transcription factor [130]. Thakur et al. reported that MMP14 can regulate the DNA damage response (DDR) by controlling integrin β1. They also reported that MMP14 inhibition elevated the sensitivity of TNBC cells to radiation therapy and doxorubicin in in vivo and in vitro settings [131], thereby improving the prognosis of patients with breast cancer. Yip et al. observed that MT4-MMP and EGFR may be co-expressed in approximately 80% of TNBCs [132]. Furthermore, Foidart et al. found that membrane-type-4 MMP (MT4-MMP), EGFR, and the retinoblastoma protein (RB) can be co-expressed in 50% of TNBCs. They also observed that this type of TNBC is highly sensitive to a combination of erlotinib and pabocitinib treatment, thereby offering a reference for the personalized treatment of TNBC [133]. In addition, in recent years, many studies have suggested that MMPs can also be used as potential biomarkers for breast cancer to help in diagnosis or to assess the efficacy of chemotherapy [134,135,136].

Zinc plays a crucial role in tumor immunity. As an ion signaling molecule, zinc is able to transmit intracellular signaling events through extracellular inputs. It also acts as a secondary messenger by mimicking cytokines and hormones. Cellular alterations exhibited by malignant cells and the rationalization of neoplastic diseases have facilitated the development of targeted therapies. Zinc transporters have been implicated as potential therapeutic targets. Compared with traditional therapies, such as chemotherapy, therapies targeting zinc transporters and using monoclonal antibodies, small-molecule inhibitors, and immunotoxins can provide effective therapeutic effects and reduce side effects. In addition, one of the characteristics of neoplastic transformation is the loss of apoptotic capacity. Zinc ions are essential for maintaining protein p53 stability and its affinity for DNA. Notably, increased concentrations of MT-1 and MT-2 lead to the removal of Zn, which leads to the destabilization and inactivation of p53, thereby inhibiting apoptosis [7]. The antioxidant, regenerative, and angiogenic functions of MT may promote cancer development, while its anti-inflammatory activity inhibits cancer.

Zinc-related metabolism plays a crucial role in the investigation and treatment of breast cancer. We summarized the role played by zinc transporters in the etiology, apoptosis, and signal transduction of breast cancer. The translation of the clinical application of zinc and related molecules in breast cancer is emphasized, including the use of serum zinc concentrations, copper/zinc ratios, and zinc isotopes to assist in the diagnosis of breast cancer, the assessment of therapeutic effects, and the identification of malignancy and subtypes. A long-term moderate intake of zinc-based diets may help to improve the prognosis of several diseases, including breast cancer. Finally, we summarized the zinc-related drug targets and determined their potential in treating breast cancer. These include the following: (1) Investigating drugs related to the expressions of zinc transporters in breast cancer cells and the induction of apoptosis. A few of these drug molecules are still under production or are being used for clinical trials. (2) Using zinc-containing nanoparticles as novel drug delivery systems to help chemotherapy or targeted immune drugs to accurately kill tumor cells and reduce systolic blood pressure. However, it must be noted that the current research and results need to be thoroughly investigated and validated before being applied in clinical settings, and a significant amount of basic and clinical research is still required to fully understand the crucial function exerted by zinc in human health. In the future, zinc sensors used to quantitatively detect zinc levels or target specific subcellular zinc repertoires could provide more accurate information about the role of abnormal zinc homeostasis in cancer development and progression. The findings of this study could offer insights into the strategies and options for the prevention, diagnosis, treatment, and efficacy assessment of breast cancer.

## 11. Conclusions

The mechanistic studies of breast cancer and zinc have been intricately linked in recent decades, and investigations into the aberrant function and activity imbalance of zinc transporters may offer novel targets and therapeutic alternatives for breast cancer. Additionally, the diagnosis, therapy, and prognosis assessment of breast cancer have all been significantly impacted by blood zinc levels, copper/zinc ratios, zinc isotopes, zinc diets, and related medications. The related mechanisms need to be explored in the future to ensure that they can be used as efficient tools for the treatment of patients with breast cancer.

## Figures and Tables

**Figure 1 nutrients-15-01703-f001:**
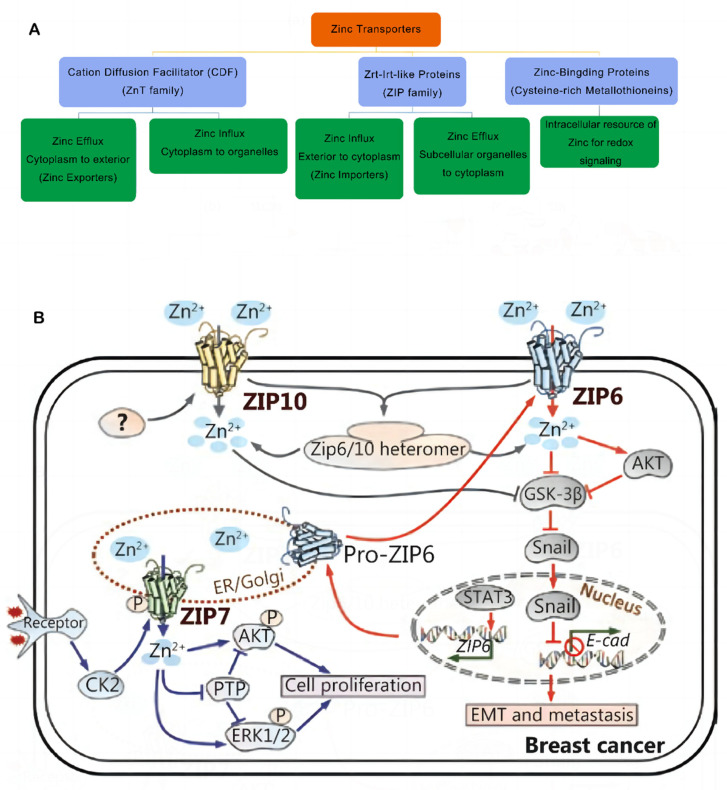
(**A**) Classification of zinc transporters and their functions. Abbreviations: ZIP, Zrt-Irt-like protein; ZnT, zinc transporter. (**B**) Zinc signaling pathways in breast cancer. The transporter-mediated imbalance of intracellular zinc can contribute to the development and progression of breast cancer. Reproduced with permission [26]. Copyright © 2022 Wiley Periodicals LLC. Reproduced with permission [10]. Copyright © 2020, Cancer Biology & Medicine.

**Figure 2 nutrients-15-01703-f002:**
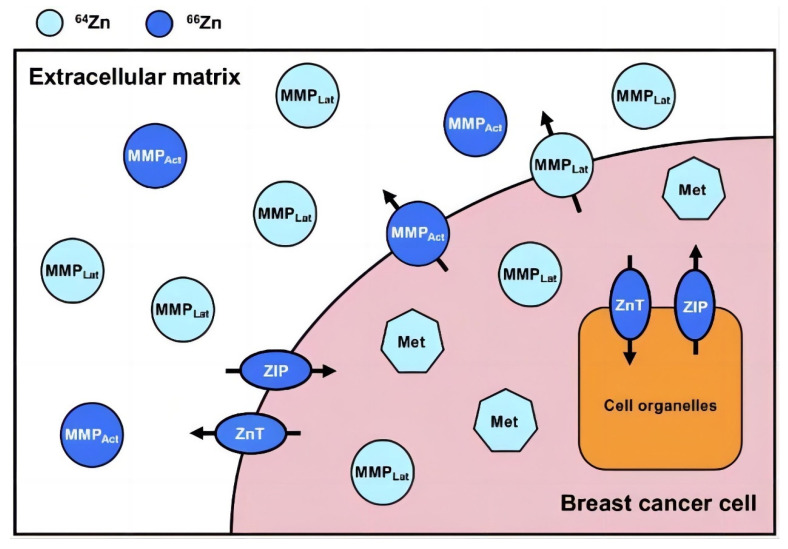
The distribution of Zn stable isotopes and their trafficking in and around a simplified breast cancer cell. Zinc in dark and light blue represents a relative enrichment in the heavy (^66^Zn) and light (^64^Zn) Zn isotope, respectively. ZIPs transport Zn into the cytoplasm both from outside the cell and from organelles. ZnTs transport Zn from the cytoplasm to both cell organelles and outside of the cell. Metallothionein and MMPs (both activated and latent) are strongly expressed compared to those in healthy breast tissue and benign tumors. Abbreviations: latent matrix metalloproteinase, MMP_Lat_ (light blue circles); activated matrix metalloproteinase, MMP_Act_ (dark blue circles); metallothionein, Met (light blue heptagons). Reproduced with permission [24]. Copyright © Oxford University Press 2021.

**Figure 3 nutrients-15-01703-f003:**
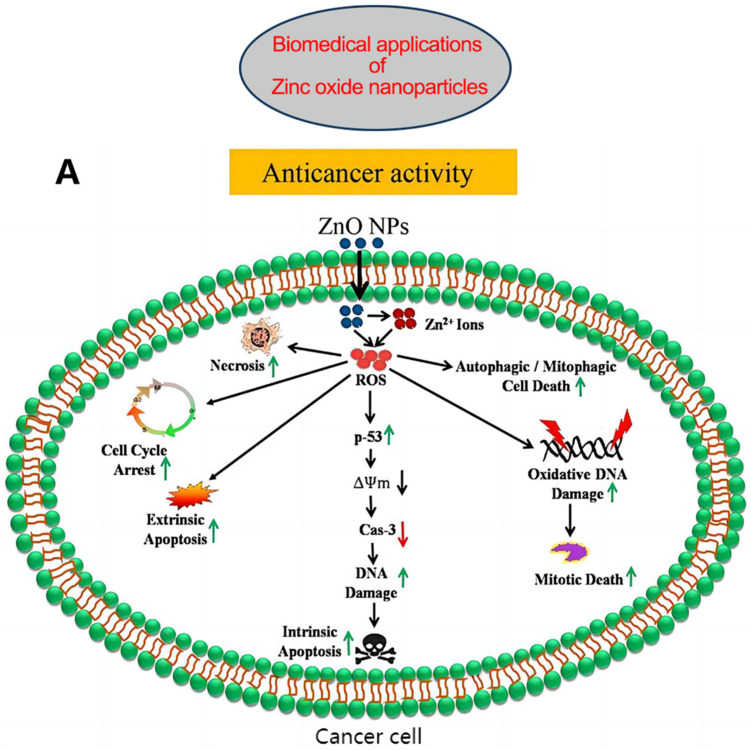
(**A**) Possible anticancer mechanisms of ZnONPs: Following internalization into cancer cells, ZnONPs produce Zn^2+^ ions via dissolution under acidic intracellular environments and cause oxidative stress. The increased ROS cause mitochondrial dysfunction and activate the intrinsic apoptosis pathway. ROS can also cause cell death via extrinsic apoptosis and necrosis pathways. ROS further induce oxidative DNA damage and mitotic death. ROS can also trigger autophagic or mitophagic cell death. (**B**) ZnONPs as anticancer drug delivery carriers; mesoporous silica nanoparticle-based DDS, where acid-sensitive ZnONPs are used as pore-blocking agents after drug loading inside channel-like pores; ZnO/polymer core–shell nanocomposite, where drugs are loaded within the hydrophobic shell; porous ZnONPs, where drugs are loaded inside the pores; ZnONP/drug complex via stable coordination bonding between the Zn^2+^ ions of ZnONPs and the oxygen-containing functional groups of the drug; porous ZnONPs loaded with drug enter the cells via endocytosis and release the drug into the cytosol after passing through the lysosome. ZnONPs also impart a synergistic anticancer effect due to the production of highly toxic Zn^2+^ ions in weakly acidic intracellular microenvironments. Abbreviations: DDS, drug delivery system; ROS, reactive oxygen species. Reproduced with permission [103]. Copyright © 2020 Elsevier B.V.

## Data Availability

The data that support the findings of this study are available from the corresponding author, Fang Y, upon reasonable request.

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
