# Peer review of "A Systematic Study on Zinc-Related Metabolism in Breast Cancer"

_nutrients, 2023, doi:10.3390/nu15071703_

Round 1
Reviewer 1 Report
Zinc is an important metal involved in many biological processes, including as a signalling mediator. Intracellular zinc levels are regulated by zinc-binding proteins and Zn transporters. ZIPs and ZnTs maintain intracellular Zn homeostasis and control important cellular functions via Zn signalling. Several studies have highlighted the important role of zinc and its transporters in cancer disease development. The authors reviewed studies on the role of zinc and its transporters in breast cancer progression.
The following are a number of observations for the authors:
Authors should write 2-3 sentences about the cellular localisation and functions of the transporters in view, as well as explain why only Zip are considered in the review.
Line 156 - The passage between Zip9 and metallothioneins should be completed, and the metallothioneins part should be enlarged.
Line 159 - the cell abbreviations need to be deciphered.
Why “Nutrients 2021,…” at the end of the pages?
Line 159 - An explanation of why ZnT suddenly appears in the Zip transporters part needs to be added.
Correct Fig 1A - too bright colours and nothing can be seen; Fig 3 - too small font, it is impossible to understand anything.
There is no reference to figures in the whole text
Why are only Zip6, 10 and 7 in Figure 1 and no 9? Metallothioneins should be added.
Lines 189-192 - The authors should discuss why treatment with zinc-supplementation is suggested for high zinc concentrations.
The authors in the review refer to papers 32, 41, 64 and 65, which are not relevant to the part of the review where they were presented, the topic and breast cancer.
In the chapter "Zinc-rich diet and breast cancer" only the last 6 lines are relevant to the topic of breast cancer, the authors should either expand this part or merge it with another.
The part on nanoparticles in the review should be made as a separate chapter.
Author Response
Dear reviewer,
Thanks very much for taking your time to review this manuscript. We really appreciate all your comments and suggestions, which have enabled us to improve our work. Based on the instructions provided in your letter, we uploaded the file of the revised manuscript. Appended to this letter is our point-by-point response to the comments raised by the reviewers. The revision instructions are as follows:
Reviewer 1
Comment 1:
Authors should write 2-3 sentences about the cellular localisation and functions of the transporters in view, as well as explain why only Zip are considered in the review.
Response:
Thanks for the kind comment. We have supplemented the original text according to your suggestions and added some literature to support the point. The expanded text is as follows:
ZnT1 is the only SLC30 member that is commonly expressed at the plasma membrane12, whereas other ZnT transporters are predominantly localized on secretory vesicles or Golgi apparatus. Most ZIP transporters localize to the plasma membrane, and their cell surface localization increases under zinc-deficient conditions, with the exception of ZIP5. ZIP and ZnT transporters work in opposite but coordinated ways to maintain cellular zinc homeostasis13. So far there is no direct structural information on any human ZnT or ZIP transporters. However, X-ray crystal and cryo-electron microscopy structures of the bacterial ZnT homologue YiiP help to reveal pathway mechanisms and structural information for zinc transport14,15. At present, there is no clear study that the tissue expression of ZnT family members is directly related to the genesis and development of breast cancer, so this paper mainly elaborates the current research status of some members of ZIP family related to breast cancer.
Comment 2:
Line 156 - The passage between Zip9 and metallothioneins should be completed, and the metallothioneins part should be enlarged.
Response:
We appreciate the reviewer’s thoughtful comments. And we totally agree that the metallothioneins part should be enlarged. In fact, we believe that the chapter on zinc transporters and breast cancer focuses on ZIP6,7,10, as these three transporters have been most closely studied in relation to breast cancer and a basic consensus is currently formed. However, other related proteins, including ZIP9 and MTs, have been less studied, and there is no strong evidence that there is an inevitable association with the pathological process of breast cancer, so they are uniformly placed in the last paragraph for a brief introduction. We supplement the article with your suggestions and add some literature. The sentences have been added to the revised manuscript.
MTs overexpression is associated with chemoresistance in patients receiving adjuvant therapy after surgery and promotes breast cancer cell invasion by increasing expression of matrix metalloproteinase-9 (MMP9)46,47. Kmiecik et al showed that MT3 may regulate the invasiveness of breast cancer cells by regulating MMP3 expression, suggesting that MT3 expression may be a potential marker of poor prognosis in triple-negative breast cancer (TNBC)48
Comment 3:
Line 159 - the cell abbreviations need to be deciphered.
Response:
We greatly appreciate your insightful comments. MDA-MB-231 and T47D cells represent different human breast cancer cell lines, differing in their p53 status, as well as estrogen receptor status, MDA-MB-231 (p53 mutant and ER-) and T47D (p53 mutant and ER+). Usually we use the abbreviations to represent the cell lines. And we added the p53 status and estrogen receptor status to explain the cell lines. Thanks for your kind advice!
Comment 4:
Why “Nutrients 2021,…” at the end of the pages?
Response:
We appreciate the reviewer’s careful comments. We did not add any headers or footers when submitting the original text. We believe that ”Nutrients 2021,…” should be modified and adjusted by the editor according to the requirements of the press, regardless of the content of this paper.
Comment 5:
Line 159 - An explanation of why ZnT suddenly appears in the Zip transporters part needs to be added.
Response:
We greatly appreciate the reviewer’s crucial suggestion and we do apologize for our mistake. We have modified the sentences and ZnT should not be appeared in this part. The change is as follows:
Chandler et al. assessed the variations occurring in the expression of Zn transporters and MTs in breast cancer cell lines and noted that MT was only overexpressed in basal-like MDA-MB-231 cells (p53 mutant and ER-) instead of T47D cells (p53 mutant and ER+). The findings suggested that subtype-specific dysregulation of zinc management could be responsible for the phenotypic properties of breast cancer.
Comment 6:
Correct Fig 1A - too bright colours and nothing can be seen; Fig 3 - too small font, it is impossible to understand anything.
Response:
We greatly appreciate the reviewer’s careful review and thoughtful comments. We have modified Figures 1 and 3 to enhance resolution as well as contrast so that readers can see the details of the figures clearly. Thanks for the reviewer’s kind suggestion.
Comment 7:
There is no reference to figures in the whole text
Response:
Thanks for your helpful comment and suggestion. We have added the references to figures in the text. And the original literature for all figures has been identified in the notes below the figures. All referenced pictures have also been granted copyright permission.
Comment 8:
Why are only Zip6, 10 and 7 in Figure 1 and no 9? Metallothioneins should be added.
Response:
We greatly appreciate your precious suggestion and comment. Like we mentioned above, in fact, we believe that the figure on zinc transporters and breast cancer focuses on ZIP6,7,10, as these three transporters have been most closely studied in relation to breast cancer and a basic consensus is currently formed. However, ZIP9 and MTs has been less studied, and there is no strong evidence that there is an inevitable association with the pathological process of breast cancer. In our literature review, no studies have shown that ZIP9-mediated apoptosis and MTs plays a key role in breast cancer genesis,such as cell proliferation, EMT and metastasis et al. Both ZIP9 and MTs tend to be biomarkers of breast cancer. So we did not add them to Figure 1B. We will also pay close attention to the subsequent research progress on ZIP9 and MTs and explore the deeper association between MTs and breast cancer. Thank you again!
Comment 9:
Lines 189-192 - The authors should discuss why treatment with zinc-supplementation is suggested for high zinc concentrations.
Response:
We appreciate the reviewer’s thoughtful and helpful comments. In line 189, Riesop 's study aimed to describe a concept along with the preceding statement, that reductions in serum zinc levels are interrelated with enrichment of zinc in breast cancer tissue. Next, Bobrowska-Korczak 's study confirmed that the biological activity of zinc in vivo depends on the size of the particles applied, as treatment with zinc microparticles had little effect on cancer progression. This indirectly demonstrates that the simply enrichment of zinc in breast cancer tissue is not able to help inhibit cancer progression, but zinc nanoparticles effectively inhibit tumor growth. We are very sorry that we did not elaborate on these two studies separately. We realized this mistake and revised and supplemented the original text, thank you!
Riesop et al. analyzed the tissue samples collected from breast cancer and healthy patients, using Laser Ablation-Inductively Coupled Plasma- Mass spectrometry technique (LA-ICP-MS) and observed that the Zn ion levels in the breast cancer tissues were 2-times higher than the control samples61. It can therefore be inferred that breast cancer tissues are enriched for zinc in blood, thereby reducing zinc levels in serum. Bobrowska-Korczak et al. showed that the dietary intake of Zn nanoparticles (ZnNPs) effectively inhibited tumor growth in breast cancer rat models. Thus, Zn supplementation could be regarded as a potential chemotherapeutic approach, which inhibited the progression of breast cancer62. Notably, rapid progression of cancer was observed in control (no additional zinc supplementation) and zinc microparticle-supplemented rats in the experiment. There was no significant difference between these two groups. However, supplementation with zinc nanoparticles strongly suppressed tumor development, revealing that the biological activity of zinc in vivo depends on the size of the applied particles instead of zinc concentration.
Comment 10:
The authors in the review refer to papers 32, 41, 64 and 65, which are not relevant to the part of the review where they were presented, the topic and breast cancer.
Response:
We appreciate the reviewer’s careful comments. Originally, we considered a brief review of the role and links of other cancers at the end of some paragraphs, and we have deleted the relevant content according to your suggestions at present, thank you very much!
Comment 11:
In the chapter "Zinc-rich diet and breast cancer" only the last 6 lines are relevant to the topic of breast cancer, the authors should either expand this part or merge it with another.
Response:
We greatly appreciate your insightful comments. Based on your comments, we expanded the basic information on dietary zinc intake, including food information and a portion of food intake, as well as the global status quo of insufficient zinc intake. We also supplemented some literature on dietary zinc intake and breast cancer risk to make the article more fully discussed. The revision of this chapter is as follows. Thanks for your valuable suggestion.
Zinc is a vital trace element that is involved in many biological processes; however, it cannot be stored in the body. Most zinc (85%) is found in the skeletal muscle and bones in the human body. A very small amount (about 0.1–1% of total) is in plasma77. Organic forms of zinc are usually better absorbed, and foods of animal origin, especially oysters, meat (beef, pork, mutton), and some seafood, nuts, seeds and dried beans are good sources of dietary zinc,. Vegetables, fruits, starchy tuberous roots and tubers have low zinc content78,79. Therefore, dietary intake of Zn metal ions is particularly important. Inadequate zinc intake can be a serious global public health problem, compromising the health of millions of adults and children80. The Lancet series on mothers and undernourished children concluded that zinc deficiency accounts for approximately 4% of child mortality and disability-adjusted life years81. Several studies have shown that dietary Zn intake may be negatively related to the risk of various diseases including depression, cognitive impairment, chronic kidney disease, and decreased immune defense7,82-84. Meanwhile, reduced dietary zinc intake may be positively associated with the progression of type 2 diabetes, atherosclerosis, and metabolic syndrome (MS)85-87. However, the correlation between zinc intake and cancer is inconclusive. A prospective study carried out in Jiangsu Province, China, with a 10-year follow-up showed that zinc intake was positively associated with all-cause mortality and cancer mortality88. In their study, Epstein et al. noted that high dietary Zn intake was associated with reduced prostate cancer-specific mortality after diagnosis89. Gutiérrez-González et al. reported that high Zn dietary intake may increase the risk of low-grade prostate cancer90. A meta-analysis showed that dietary zinc intake significantly reduced the risk of pancreatic cancer in the US population91. Luo et al. conducted a case-control study and observed that the intake of Zn ions was not linked to colorectal cancer risk92. A German study investigated the relationship between breast cancer incidence and intake of vegetables, fruits, and micronutrients and found that high zinc intake may reduce breast cancer risk93. Furthermore, no clear relationship was noted between dietary Zn intake and the overall breast cancer risk or the risk of various breast cancer subgroups94. However, in terms of the survival rate, Bengtsson et al. observed that the moderate/high Zn intake could help in increasing the survival rate of breast cancer patients, based on the Malmö Diet and Cancer Study in Sweden25. Pan and colleagues from Canada studied 2322 patients to assess the association between antioxidants and breast cancer risk and found supplementation of 10 years or longer was associated with statistically significant reduction in premenopausal breast cancer risk for zinc, though the overall effect of total dose or intake from both diet and supplement cannot be determined95. While a cohort study from Spain involving 9983 participants reported that there was no evidence for a consistent association between intake of zinc and breast cancer risk either in overall or premenopausal women96. The correlation between dietary Zn intake and breast cancer morbidity and mortality still needs to be corroborated by further research.
Comment 12:
The part on nanoparticles in the review should be made as a separate chapter.
Response:
Thanks the reviewer for pointing this out. We totally agree with this suggestion. We have changed the part on nanoparticles into a separate chapter. Thanks!

Reviewer 2 Report
The present study provides a very elucidative review of the mechanisms involved in the relationship between zinc and breast cancer, congratulations to the authors.
However, I would like to point out some modifications that would be necessary to broaden the understanding and organization of the manuscript.
General comments:
- Add a topic of methodology, which describes how the research was performed, what databases, keywords and how did the included articles was selected (was there cut-off per year? were only human cell studies included or animal studies as well? various types of cancer or only breast cancer?)
- Evaluate the repetition of some sentences throughout the text in order to make it more objective and attractive to the reader
-Improve the resolution of images, such as Figure 1A
- Quote the images in the body of the text
Comments per section:
1) Introduction:
* Ln 80: "recently, several authors..." . It would be important to refer to the articles that led to construction of this phrase.
2) Zinc transporters and breast cancer:
*Ln 93: "Therefore, it has been highlighted that high ZIP6 expression is a reliable marker of luminal A subtypes in breast cancer". A phrase very similar to this one is on line 89. To make the writing more direct, I recommend taking it off.
3) Zinc-rich diet and breast cancer:
Bring more information about food intake of zinc. Did the studies evaluated food intake or supplemented as well? Are there studies on the impact of supplementation?
4) Discussion
The first two paragraphs have a characteristic of explanation of mechanisms, and are not the most appropriate to start the discussion. Consider starting the discussion by making clear the main contributions of the article to science and clinical practice. In addition, it would be important to highlight, throughout the discussion, what are the main gaps in this area and bring recommendations for future studies.
Author Response
Dear reviewer,
Thanks very much for taking your time to review this manuscript. We really appreciate all your comments and suggestions, which have enabled us to improve our work. Based on the instructions provided in your letter, we uploaded the file of the revised manuscript. Appended to this letter is our point-by-point response to the comments raised by the reviewers. The revision instructions are as follows:
Reviewer 2
Comment 1:
Add a topic of methodology, which describes how the research was performed, what databases, keywords and how did the included articles was selected (was there cut-off per year? were only human cell studies included or animal studies as well? various types of cancer or only breast cancer?)
Response:
We greatly appreciate the reviewer’s thoughtful and insightful comments. We have added the topic of methodology accordingly. The revision of methodology is as follows. Thanks for your valuable suggestion.
To describe the limitations of existing research, suggest avenues for future research and therapeutic strategies to address gaps in the current literature and provide practical recommendations for Zinc-Related Metabolism in Breast Cancer. For this study, a search of the literature was conducted to identify reviews, randomized controlled trials, systematic reviews, meta analysis and articles which investigated zinc and breast cancer (e.g. zinc homeostasis, zinc transporters, zinc isotopes, zinc-related treatment and targets for breast cancer ). The electronic databases PubMed, Web of Science and Google Scholar were searched for studies published from January 1995 to February 2023. Types of studies include human clinical studies, human cell studies, and animal experimental studies. The research direction is mainly aimed at breast cancer, but it also covers some research on other cancers including lung cancer, liver cancer, and thyroid cancer.
Comment 2:
Evaluate the repetition of some sentences throughout the text in order to make it more objective and attractive to the reader
Response:
We thank the reviewer for this valuable suggestion. The whole manuscript has been polished accordingly, and we removed some repeated sentences to make the text more succinct. Therefore, the readers may understand our work more clearly.
Comment 3:
Improve the resolution of images, such as Figure 1A
Response:
We greatly appreciate the reviewer’s careful review and thoughtful comments. We have modified Figures 1 and 3 to enhance resolution as well as contrast so that readers can see the details of the figures clearly. Thanks for the reviewer’s kind suggestion.
Comment 4:
Quote the images in the body of the text
Response:
Thanks for your helpful comment and suggestion. We have quoted the figures in the text. And the original literature for all figures has been identified in the notes below the figures. All referenced pictures have also been granted copyright permission.
Comment 5:
Introduction: * Ln 80: "recently, several authors..." . It would be important to refer to the articles that led to construction of this phrase.
Response:
We greatly appreciate the reviewer’s crucial suggestion and we do apologize for our mistake. We have added several references to support our argument. We’re sorry that we forgot to refer to the phrase. Thank you again!
Comment 6:
Zinc transporters and breast cancer: *Ln 93: "Therefore, it has been highlighted that high ZIP6 expression is a reliable marker of luminal A subtypes in breast cancer". A phrase very similar to this one is on line 89. To make the writing more direct, I recommend taking it off.
Response:
We greatly appreciate your insightful and careful comments. We do found that these two phrases are very similar and express the same meaning. So we deleted the sentence on line 89 to make the manuscript more succinct. Thanks for your kind suggestion.
Comment 7:
Zinc-rich diet and breast cancer: Bring more information about food intake of zinc. Did the studies evaluated food intake or supplemented as well? Are there studies on the impact of supplementation?
Response:
We greatly appreciate your insightful comments. Based on your comments, we expanded the basic information on dietary zinc intake, including food information and a portion of food intake, as well as the global status quo of insufficient zinc intake. We also supplemented some literature on dietary zinc intake and breast cancer risk to make the article more fully discussed. The revision of this chapter is as follows. Thanks for your valuable suggestion.
Dietary Zinc intake and breast cancer:
Zinc is a vital trace element that is involved in many biological processes; however, it cannot be stored in the body. Most zinc (85%) is found in the skeletal muscle and bones in the human body. A very small amount (about 0.1–1% of total) is in plasma77. Organic forms of zinc are usually better absorbed, and foods of animal origin, especially oysters, meat (beef, pork, mutton), and some seafood, nuts, seeds and dried beans are good sources of dietary zinc,. Vegetables, fruits, starchy tuberous roots and tubers have low zinc content78,79. Therefore, dietary intake of Zn metal ions is particularly important. Inadequate zinc intake can be a serious global public health problem, compromising the health of millions of adults and children80. The Lancet series on mothers and undernourished children concluded that zinc deficiency accounts for approximately 4% of child mortality and disability-adjusted life years81. Several studies have shown that dietary Zn intake may be negatively related to the risk of various diseases including depression, cognitive impairment, chronic kidney disease, and decreased immune defense7,82-84. Meanwhile, reduced dietary zinc intake may be positively associated with the progression of type 2 diabetes, atherosclerosis, and metabolic syndrome (MS)85-87. However, the correlation between zinc intake and cancer is inconclusive. A prospective study carried out in Jiangsu Province, China, with a 10-year follow-up showed that zinc intake was positively associated with all-cause mortality and cancer mortality88. In their study, Epstein et al. noted that high dietary Zn intake was associated with reduced prostate cancer-specific mortality after diagnosis89. Gutiérrez-González et al. reported that high Zn dietary intake may increase the risk of low-grade prostate cancer90. A meta-analysis showed that dietary zinc intake significantly reduced the risk of pancreatic cancer in the US population91. Luo et al. conducted a case-control study and observed that the intake of Zn ions was not linked to colorectal cancer risk92. A German study investigated the relationship between breast cancer incidence and intake of vegetables, fruits, and micronutrients and found that high zinc intake may reduce breast cancer risk93. Furthermore, no clear relationship was noted between dietary Zn intake and the overall breast cancer risk or the risk of various breast cancer subgroups94. However, in terms of the survival rate, Bengtsson et al. observed that the moderate/high Zn intake could help in increasing the survival rate of breast cancer patients, based on the Malmö Diet and Cancer Study in Sweden25. Pan and colleagues from Canada studied 2322 patients to assess the association between antioxidants and breast cancer risk and found supplementation of 10 years or longer was associated with statistically significant reduction in premenopausal breast cancer risk for zinc, though the overall effect of total dose or intake from both diet and supplement cannot be determined95. While a cohort study from Spain involving 9983 participants reported that there was no evidence for a consistent association between intake of zinc and breast cancer risk either in overall or premenopausal women96. The correlation between dietary Zn intake and breast cancer morbidity and mortality still needs to be corroborated by further research.
Comment 8:
Discussion: The first two paragraphs have a characteristic of explanation of mechanisms, and are not the most appropriate to start the discussion. Consider starting the discussion by making clear the main contributions of the article to science and clinical practice. In addition, it would be important to highlight, throughout the discussion, what are the main gaps in this area and bring recommendations for future studies.
Response:
We greatly appreciate your insightful comments. Based on the reviewer’s valuable suggestions and insightful comments. We focused on modifying the discussion section. We deleted the repeated statements in the previous version, synthesize the existing literature and studies to give the corresponding conclusions and judgments, and give some suggestions on the comprehensive treatment of zinc and breast cancer in the future. (Line 389, 404, 428, 449 and 480)

Reviewer 3 Report
The opinion of the reviewer about the work of Qu et al. concerning the role zinc in breast cancer could be divided into two parts - A) scientific and B) overall reader´ s.
A) Authors attempted to summarize evidence concerning various ways zinc and its intracellular management might be involved in breast cancer including zinc-related metabolism. Listed are zinc transporters, however other mechanisms and players including various zinc buffers and mufflers are not even mentioned. Furthermore, the role of free zinc pools versus bound zinc are not touched at all despite the crucial significance of these two zinc forms in various cell functions including carcinogenesis.
Other zinc biology aspects covered seem unbalanced and above all misleadingly mislabeled. This is for instance the case of the header "zinc isotopes and breast cancer" which may imply zinc isotopes play important roles in biology of breast cancer rather than the fact that zinc isotopes are used for detection of changes in zinc-related metabolism. Here a different approach should be used to make the intended information delivered proper way.
Discussion is not really discussion, it is listing of many reports dealing with zinc in relation to cancer/breast cancer without so much as attempting to synthesize, discuss and/or evaluate the presented information. Instead authors present numerous phrases and statements which they keep repeating throughout the entire manuscript without offering some credible explanations.
That leads to an overall impression as per part B). The text is not coherent, mixes various approaches and levels and does not offer any scientifically robust conclusions. Together with a number of language mistakes - sometimes factual " histological malignancy" - page 2, use of non-scientific expressions "a lot of" along with some sentences not making the sense results is difficult, confusing and not particularly valuable text.
it is believed that the role of zinc in various cancer, in particular, breast cancer is interesting and robust topic. Still, the presents manuscript as well as many other similar works simply fail to provide (or at least suggest) solutions for fundamental questions in this field; i.e is zinc-related intracellular management merely a result of malignant transformation and cells just mismanage zinc among other elements and players to acquire survival advantages? Or is zinc one of the centerpoints of malignant transformation similar to the case of prostate cancer? It is expected that the present review will address these issues rather than listing reported evidence and repeating phrase just like ones in the conclusions section?
Author Response
Dear reviewer,
We appreciate the reviewer for taking the time to carefully review the manuscript and give detailed
and constructive comments, which has greatly helped to improve this paper. We thoroughly reviewed our manuscript against your comments and suggestions and carefully revised it from two aspects: scientificity and overall readers.
First of all, we strongly agree with the evidence that participants other than zinc transporters should be added, and we supplement this aspect in multiple parts of the article. We systematically searched relevant articles on mechanisms of cellular zinc homeostasis and intracellular management including zinc buffering and muffling. We address the role of free zinc pools, bound zinc, and associated zinc homeostasis. Chemicals or treatments that specifically modulate zinc transporter function or zinc levels in subcellular zinc stores may serve as effective tools for treating cancer patients. Further studies are needed to elucidate the exact role of zinc in breast cancers. Associated studies are integrated in the discussion as well as in other sections. (Line 51, 180, and 389)
Secondly, we modified the content of misleadingly mislabeled other aspects of zinc biology by deleting the title "Zinc Isotopes and Breast Cancer" and replacing it with "Zinc isotopes in breast cancer detection and prognosis prediction". We also deleted the title "Zinc-rich diet and breast cancer" and replaced it with "Dietary Zinc intake and breast cancer" to convey more accurate and scientific information to readers. (Line 241 and 279)
Based on the reviewer’s valuable suggestions and insightful comments. We focused on modifying the discussion section. We deleted the repeated statements in the previous version, synthesize the existing literature and studies to give the corresponding conclusions and judgments, and give some suggestions on the comprehensive treatment of zinc and breast cancer in the future. (Line 389, 404, 428, 449, and 480)
Regarding part B, we have corrected the text inconsistency by modifying several grammatical errors that include what you mentioned. Some other grammatical errors in the article have been removed. At the same time, the article has been polished and modified under the review of professional English speakers, and we also provide the certificate of English polishing in this response.(Line 76, 98, 193, 244, 281, 311, 413, and 422)
To sum up, this modification addressed the content of zinc-related intracellular management from multiple aspects and complements other mechanisms and players. Combined with the existing literature and studies, it cannot be proved that zinc is one of the central points of malignant transformation or zinc dominates the process of oncogenesis. The aim of this paper is to describe the role and advantages of zinc and related metabolites in the occurrence, development, diagnosis, comprehensive treatment, and prognostic evaluation of breast cancer. The specific role of zinc in the tumor immune microenvironment of breast cancer and the specific mechanisms involved in tumor formation still need further in-depth study. Our team will also subsequently conduct many aspects of research such as basic experiments and clinical trials for related mechanisms.
Thanks again for the reviewer’s hard work!

Round 2
Reviewer 1 Report
In the Comment 6 I asked to enlarge not only the resolution but also the font on the figure. I continue to see too small font, it is impossible to understand this figure. Authors should reorganise this figure
In the Comment 12 – I don’t see any difference in the old and new versions of the paper. Authors answered that they put nanoparticles as separate chapter, I don’t see!
Comment 10: The authors in the review refer to papers 32, 41, 64 and 65, which are not relevant to the part of the review where they were presented, the topic and breast cancer. For this comment authors declared that “we have deleted the relevant content according to your suggestions at present”, HOWEVER I found these references in the text!!!!! For example, 64 and 65 become 71 and 72
I would like to point out that the response to the reviewer is not only "Thank you very much for the comments", but also the actual corrections in the text of the paper.
The reviewer is not obliged to check the previous text of the paper against the new one in order to check the veracity of the corrections!!!
Author Response
Dear reviewer,
We are very sorry for our team's colossal mistake. In the first round of revision, we responded to the comments of the three reviewers respectively. Problems arose during the integration of the changes to the final three versions, to the extent that responses to these comments were not integrated together. This kind of problem should have been completely avoided.
We reorganized Figure 3 and enlarged the font so that readers could clearly see the text in the picture. We removed papers 32, 41, 64, and 65 from previous editions. In addition, we split the part about nanoparticles from the original paragraph into a chapter independently, and this chapter is titled Zinc oxide nanoparticles and breast cancer.
We greatly appreciate your valuable suggestions and criticisms. We must pay attention to the rigor of the revision in the future and we do apologize for our mistakes. Thank you again!

Reviewer 3 Report
Authors addressed concerns and modified their manuscript accordingly.
Author Response
Thanks very much for taking your time to review this manuscript. We really appreciate all your comments and suggestions, which have enabled us to improve our work.